# Structural Topology Optimization of Reflective Mirror Based on Objective of Wavefront Aberration

Wenli Li [1,2,3], Yincheng Shi [1,2,3], Chong Wang [1,3], Yufeng Tang [1,2,3], Yi Yu [1,3] and Zhenyu Liu [1,3,*]

1    Changchun Institute of Optics, Fine Mechanics and Physics (CIOMP), Chinese Academy of Sciences, Changchun 130033, China
2    University of Chinese Academy of Sciences, Beijing 100049, China
3    Key Laboratory of Space-Based Dynamic & Rapid Optical Imaging Technology, Chinese Academy of Sciences, Changchun 130033, China
*    Correspondence: liuzy@ciomp.ac.cn

**Abstract:** Due to the increasing requirements for the imaging quality of optical systems, the design method for optical–mechanical structures has become a research hotspot in recent decades. To improve the imaging performance of the reflective system, it is often necessary to increase the aperture of the mirror. To meet the imaging quality and lightweightedness requirements of the mirror, the topology optimization method aiming at the minimization of wavefront aberration is proposed. The optical–mechanical coupling relationship is established by ray tracing of the deformed mirror surface fitted by orthogonal bases. The topology optimization model is established by the solid isotropic material with penalization model (SIMP). Additionally, the adjoint method is used to analyze the sensitivity of the objective and the constraints. To illustrate the rationality and effectiveness of the method, the mirrors of the Cassegrain system have been optimized under the action of gravity with the objective of the weighted sum of squares of wavefront aberration coefficients under the constraints of the mass of the design domain, the rigid body displacement of the mirror surface, and the residual of deformation fitting. The results show that the proposed method can effectively improve imaging performance under the condition of satisfying the constraints. In addition, the optimization method with wavefront aberration as the objective is a concrete application of the idea of opto-mechanical integration, which can improve optical performance more directly and effectively.

**Keywords:** mirror structure; zernike fitting; ray tracing; optical path difference; wavefront aberration; topology optimization; cassegrain system

## 1. Introduction

The mirror is an important part of the modern telescopic system, and the accuracy of its position and surface shape directly affects the imaging quality of the system. For an ideal optical system, the imaging quality is limited by the diffraction limit of the optical mirror. According to the calculation method of the diffraction limit of the optical system, the aperture of the mirror should be increased to improve the imaging quality of the system [1]. However, increasing the optical aperture might increase the deformation of the mirror surface caused by gravity, which will reduce the imaging quality. In addition, the increase of the mass will reduce the maneuverability of the system. Therefore, the lightweight design method of the mirror is of great significance to improve the imaging quality, and has become the focus of the opto-mechanical structure design.

The traditional methods mainly include scheme comparison and parameter optimization. The back-contour form, open form and lightweight hole shape of the mirror are determined by scheme comparison. The shapes of lightweight holes are mainly triangular, quadrilateral, hexagonal, fan-shaped and circular. Yan et al. [2] and Guo et al. [3] compared the structural stiffness, manufacturability and lightweight rate of mirrors with

different lightweight holes. Valente et al. [4] analyzed and compared open-back, symmetric sandwich and contoured back mirrors from the aspects of weight, deformation and manufacturability. The materials at different positions of the mirror have different contributions to the stiffness, and the lightweight rate can be improved by removing the materials with low contributions to the stiffness. Chen et al. [5] analyzed the relationship between the geometric parameters of the honeycomb sandwich mirror and the structural stiffness and surface shape accuracy by theoretical analysis and finite element method. Kihm et al. [6] used the CATIA to conduct parametrization and finite element analysis on a $\phi$1 m glass–ceramic mirrors, and used a genetic algorithm to find the optimal solution. Zhang et al. [7] used a genetic algorithm to optimize the parameters of the circular mirror with hexagonal lightweight holes. Chen et al. [8] parameterized a glass–ceramic mirror with a diameter of 566 mm, used the ANSYS to solve the displacement of mirror nodes, and used the MATLAB program to perform Zernike fitting on the deformed mirror nodes. Finally, the SmartDo software was used to find the optimal solution. Wang et al. [9], based on the back propagation neural network, completed the structural optimization design of the $\phi$2.8 m aperture mirror. Liu et al. [10] took the weighted compliance of multiple working conditions as the objective and carried out topology optimization design of the triangular lightweight mirror under the constraints of mirror nodes displacement, mass and symmetry. The mirror parameters were classified according to the results, then the parameters were optimized based on the ISIGHT platform. Due to the limitation of the original structure, the performance improvement of the mirror is limited by parameter optimization. Because the topology of the original structure is unchanged in parameter optimization, the performance improvement of the mirror is limited. In addition, the initial structure of the mirror depends heavily on the experience of the designer. Therefore, the mirror structure has a lot of room for improvement in terms of lightweightedness and surface accuracy.

Topology optimization is a creative design method in the field of structural design, which can achieve the optimal distribution of materials under given constraints to optimize the objective. Topology optimization, which has become a hot spot in the field of structural design, breaks the norms of traditional structural design in order to obtain novel and better structures. Commonly used topology optimization methods include the homogenization approach [11], SIMP [12,13] and level set [14]. There have been many research papers exploring applications of topology optimization methods in the structural design of mirrors. Park et al. [15,16] used the RMS of the normal displacement of the mirror as the objective to design the structure of the mirror under the volume constraint. Liu et al. [17] optimized the rib layout and height of a $\phi$2 m aperture mirror with compliance as the objective under the volume constraint. Fan et al. [18] completed the optimization design of a $\phi$260 mm aperture open-back mirror in the same way, used 3D printing technology to complete the manufacture of the mirror and carried out experimental verification. Li et al. [19] optimized a $\phi$760 mm SIC mirror with compliance as the objective. According to the results, the mirror meeting the engineering requirements was designed and the parameters were optimized. Qu et al. [20] completed the structural optimization of the mirror with a center hole under the constraints of the volume and the nodal displacement of the mirror surface, with the objective of multi-case weighted compliance. In addition, Qu et al. [21] carried out the topology optimization design of a rectangular mirror with mass as the objective under the constraints of fundamental frequency and mirror nodes displacement, and completed the parametric optimization and performance analysis according to the optimized structure. Wang et al. [22] completed the optimization of a $\phi$700 mm aperture mirror under the constraints of volume and PV with the frequency as the objective, and analyzed the mechanical and thermal properties. RMS and PV reflect the overall quality of the mirror, without optical characteristic. The specific mirror deformation cannot be determined. There is no clear functional relationship between the stiffness and the imaging quality. The structure with maximum stiffness is not necessarily the mirror structure with

the best imaging quality. At the same time, specific aberrations cannot be optimized directly and efficiently based on these objectives and constraints.

To obtain a high lightweight rate, high imaging quality, stable and reliable mirror, it is necessary to use the idea of optical–mechanical integration to carry out the research of mirror structure design. Shi et al. [23] used the Zernike bases to fit the z-direction deformation of the mirror surface by integrating on the mirror surface, established a topological optimization model about the fitting coefficients and completed the support structure optimization of the mirror. This method can describe the deformation of the mirror surface in detail. Based on their work, the topology optimization method with the objective of wavefront aberration is studied. The functional relationship between optical aberration and mirror structure can be established by using a deformed mirror surface for optical analysis, which makes it realizable to optimize the topology of mirror structure based on the objective of wavefront aberration. In the Section 2, the topology optimization method of reflective mirror is introduced in detail, including the sag displacement of deformed mirror surface, the rigid body displacement fitting, Zernike fitting, ray tracing, topology optimization model, sensitivity analysis by adjoint method, and topology optimization process. The primary and secondary mirrors of the Cassegrain system are optimized in the Section 3. The optimization results are analyzed in the Section 4, and the conclusions are introduced in the Section 5.

In this paper, the topology optimization method aiming at the minimization of wavefront aberration is proposed. The analytical expression of the deformed mirror surface is obtained by fitting the mirror deformation with the orthogonal polynomials under the finite element framework. To integrate opto-mechanical coupling into topology optimization, the deformed mirror surface is used for ray tracing and the wavefront aberration is analyzed. The topology optimization model based on the objective of wavefront aberration is established by using the SIMP method. The adjoint method is used to analyze the sensitivity of the objective and constraints to the design variables, which overcomes the problem of excessive calculation in the sensitivity analysis of the difference method. A mathematical model is established for the mirrors of the Cassegrain telescopic system, which takes the weighted sum of squares of Zernike coefficients of wavefront aberration as the objective, and takes the mass, the residual of mirror deformation fitting, and the mirror surface rigid body displacement as the constraints. The structural optimization design of a $\phi$450 mm annular primary mirror in the vertical and horizontal conditions of the optical axis and a $\phi$105 mm circular secondary mirror in the vertical condition of the optical axis has been completed by using the COMSOL software. By analyzing the optimized structures, the aberration objective of the primary mirror is reduced by 93.6% in the vertical optical axis and 34.3% in the horizontal optical axis. Additionally, the aberration objective of the secondary mirror in the vertical optical axis is reduced by 85.6%. At the same time, the weighted sum of squares of wavefront aberration coefficients of the Cassegrain system composed of the optimized structures is reduced by 93.9%. The results show that the surface accuracy and imaging quality of the mirror can be effectively improved under the constraints, and the method in this paper is effective and feasible for different working conditions and different types of mirrors.

## 2. Topology Optimization Method of Reflective Mirror

The structure of the mirror has a crucial impact on its lightweightedness, surface accuracy and imaging quality. The improvement of lightweightedness will reduce the stiffness of the mirror and make the mirror deformation more sensitive to the load. Faced with this contradictory performance, topology optimization could consider all aspects of the requirements, as far as necessary to improve the performance of the mirror. Under the action of external load, the mirror surface is deformed, and the deformed surface causes the change of ray trajectory, which, in turn, affects the imaging quality of the mirror. The governing equation of static deformation is shown in Equation (1). $\mathbf{c}_\varepsilon$ is the elasticity tensor, $\mathbf{u}$ is the displacement vector, and $\mathbf{f}$ is the load vector. The mirror deforms under

the action of external load. $\Gamma$ represents the ideal mirror surface. The deformed mirror surface is related to the deformation displacement $\mathbf{u}$ and the original surface $\Gamma$, which can usually be expressed as $\Gamma'(\Gamma, \mathbf{u})$. The governing equation is difficult to obtain an analytical solution with the specified boundary conditions. Some numerical methods are usually used for analysis, such as the finite element method. Equation (2) can be obtained by solving Equation (1) with the finite element method. Therefore, it is difficult to directly obtain the analytical expression of the deformed surface, which increases the difficulty of analysis and optimization. In this paper, the structural optimization method of the mirror is studied under the finite element framework.

$$- \nabla \cdot (\mathbf{c}_\varepsilon \nabla \mathbf{u}) = \mathbf{f} \tag{1}$$

$$\mathbf{KU = F} \tag{2}$$

To improve the performance of the mirror, this paper uses the Zernike polynomials to fit the mirror deformation and combines it with topology optimization from the perspective of optical–mechanical coupling. The topology optimization model is constructed by SIMP. Through the functional relationship between design variables, elastic modulus, and wavefront aberration structural optimization with wavefront aberration as the objective can be achieved.

### 2.1. The Sag Displacement of Deformed Mirror Surface

To improve the description accuracy of the surface deformation, the influence of nodes displacement in three directions is considered. Compared with the method that only considers the displacement of nodes in the Z direction, the description of mirror deformation is more accurate. Firstly, the projection of the deformed mirror surface in the optical axis direction is established. Secondly, under the finite element system, the sag displacement inside the projective plane is calculated. Then, the Zernike bases are used to fit the deformation.

The ideal mirror surface in engineering usually has analytical expression, $z = f(x, y)$. As shown in Figure 1, under the action of external force, the displacement of the mirror surface node A is $(d_{x_i}, d_{y_i}, d_{z_i})$. The position of the node A after deformation is B. Its position on the projection plane is D, and the sag displacement of deformed mirror surface to be fitted is $\delta_i = z_i + d_{z_i} - f(x_i + d_{x_i}, y_i + d_{y_i})$. After the mirror is discretized by elements, the projection of the mirror surface and the fitted deformation amount of the projection nodes can be obtained according to the coordinates and displacements of the grid nodes.

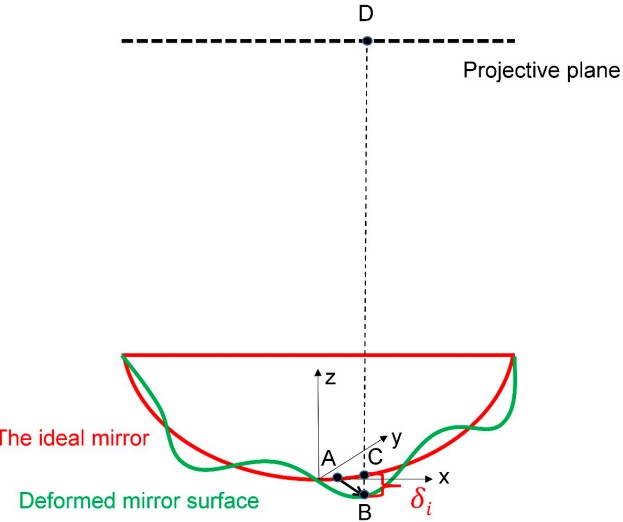

**Figure 1.** The schematic representation of the projection of deformed mirror surface and the sag displacement of the projection node. A: $(x_i, y_i, z_i)$; B: $(x_i + d_{x_i}, y_i + d_{y_i}, z_i + d_{z_i})$; C: $(x_i + d_{x_i}, y_i + d_{y_i}, f(x_i + d_{x_i}, y_i + d_{y_i}))$; D: $(x_i + d_{x_i}, y_i + d_{y_i}, z_t)$.

### 2.2. Zernike Fitting of Mirror Surface Deformation

Fitting the deformation of the mirror surface with orthogonal analytical bases can relatively accurately describe the deformed surface, which makes the discrete deformation data simple. The wavefront aberration can be obtained by ray tracing based on the fitted surface to evaluate the imaging quality, which makes the topology optimization of optical–mechanical integrated possible. It should be noted that the orthogonal bases need to be determined according to the shape of the mirror surface or the shape of the wavefront. The primary mirror of the Cassegrain system is annular, and its deformation is fitted by the annular Zernike polynomials. The secondary mirror is circular, and the fringe Zernike polynomials are used. The fitting in this paper is carried out under the finite element framework, which is illustrated by the fringe Zernike fitting of mirror surface deformation on discretized points.

According to the literature [24], the fringe Zernike polynomials are orthogonal on the unit circle. The fitting coefficients of orthogonal bases are independent of each other. The deformation of the mirror surface can be described as a weighted sum of the displacement modes represented by polynomials. The coefficients are calculated by the least square method. The quadratic error function is as follows:

$$E = \sum_{i=1}^{q} w_{1i} \left( \delta_i - \sum_{j=1}^{\widetilde{a}} C_{1j} \varphi_{1ji} \left( \widetilde{r}, \widetilde{\theta} \right) \right)^2 \tag{3}$$

where $q$ is the number of nodes to be fitted, $w_{1i}$ is the weight of the node $i$ in the projection plane, $\delta_i$ is the deformation of the node $i$ to be fitted, $\widetilde{a}$ is the number of bases, $C_{1j}$ is the fitting coefficient of the $j$th basis, $\varphi_{1ji}$ is the value of the $j$th term of the bases used to fit the mirror surface deformation at the node $i$, $\widetilde{r}$ is the polar radius, $\widetilde{\theta}$ is the polar angle. To ensure the orthogonality of the bases and improve the accuracy of the fitting, the deformed mirror surface is projected along the optical axis and fitted on the projection plane. To get the coefficients, the error function $E$ needs to be minimized. Therefore, the following equation can be obtained.

$$\mathbf{H}_1 \mathbf{C}_1 = \mathbf{P}_1 \tag{4}$$

where $\mathbf{H}_1$ is the coefficient matrix, $\mathbf{C}_1$ is the vector of fitting coefficients to be solved, and $\mathbf{P}_1$ is the right–hand term.

### 2.3. The Rigid Body Displacement Fitting of Mirror Surface

The fitting with orthogonal bases is for a single vector direction, so it cannot represent the rigid motion of the optical surface in the direction of 6 degrees of freedom. The Zernike fitting of the mirror deformation cannot obtain information about the lateral displacement of the surface or the rotation around the optical axis. To fully describe the mirror deformation, it is necessary to fit the rigid displacement of the mirror. For the optical system without active adjustment, to maintain the stability of the imaging quality, the rigid body displacement of the mirror surface should be constrained perfectly to reduce the influence on the imaging quality.

The rigid body displacement of the mirror surface is calculated by the least square method [24]. For a mirror discretized by elements, the rigid motion of the surface can be calculated by the area-weighted average motion. The translations of the mirror surface in three directions are $T_x$, $T_y$, $T_z$, and the rotations around x, y, and z axes are $R_x$, $R_y$, $R_z$ respectively. The rigid body displacement at the node A is $d_{\widetilde{x}_i}, d_{\widetilde{y}_i}, d_{\widetilde{z}_i}$.

$$\begin{aligned}
d_{\widetilde{x}_i} &= T_x + z_i R_y - y_i R_z \\
d_{\widetilde{y}_i} &= T_y - z_i R_x + x_i R_z \\
d_{\widetilde{z}_i} &= T_z + y_i R_x - x_i R_y
\end{aligned} \tag{5}$$

The quadratic error function of rigid body displacement is as follows.

$$E' = \sum_{i=1}^{q} \mathrm{m}_i \left[ \left(d_{x_i} - d_{\widetilde{x}_i}\right)^2 + \left(d_{y_i} - d_{\widetilde{y}_i}\right)^2 + \left(d_{z_i} - d_{\widetilde{z}_i}\right)^2 \right] \tag{6}$$

where $q$ is the number of the mirror surface nodes, and $m_i$ is the weight of the node $i$. By taking the partial derivatives of each term and setting them equal to 0, one can obtain the equation about the rigid body displacement as follows.

$$\mathbf{DT = G} \tag{7}$$

where $\mathbf{T} = [T_x, T_y, T_z, R_x, R_y, R_z]^T$, $\mathbf{D}$ is the coefficient matrix, and $\mathbf{G}$ is the right–hand term.

*2.4. Ray Tracing*

The optical–mechanical coupling effect is generated on the mirror surface, and the deformation of the mirror leads to the change of the light propagation path, which affects the imaging quality. The influence of the light energy on the mirror temperature is neglected, so the optical–mechanical coupling studied here is unidirectional. To apply the concept of opto-mechanical integration to the topology optimization of mirror structure, the mathematical model of topology optimization should be constructed by establishing the clear functional relationship between aberration, mirror deformation and design variables through ray tracing. Therefore, ray tracing is performed based on the fitted mirror deformation. It can reflect the influence of mirror deformation on light propagation to the greatest extent, rather than only reflect the influence of several rigid body displacements. At the same time, the functional relationship between aberration and design variables can be established, which provides convenience for the solution of sensitivity and makes the topology optimization of opto-mechanical coupling possible.

The propagation of light in space satisfies the differential Equation (8) [25].

$$\frac{\mathrm{d}}{d\widetilde{s}} \left( \mathbf{n} \frac{d\mathbf{r}}{d\widetilde{s}} \right) = grad\mathbf{n} \tag{8}$$

where $\widetilde{s}$ is the arc length of the ray, $\mathbf{r}$ is the position vector of the ray, and $\mathbf{n}$ is the refractive index, which is the quantity field of spatial distribution. When the space medium is uniformly distributed, the trajectory of the ray is straight. In a space coordinate system, $\mathbf{a}$ is the unit vector of the direction in which the ray travels. $\mathbf{b}$ is the position vector of the ray's exit point, and $t$ is the optical path. The propagation equation of light can be expressed as $\mathbf{r} = t\mathbf{a} + \mathbf{b}$.

Ray tracing is a method to calculate the propagation path of light, that is, the intersections with the various optical surfaces, including the deformed mirror surface, the Gaussian reference sphere, and the Gaussian image plane. The accuracy of ray tracing is directly related to the calculation of aberration. The fitted mirror surface has an analytical expression, and the intersection point can be accurately solved by the numerical method. The normal at the intersection can be easily obtained according to the expression, which provides convenience for tracing the reflected light.

The expression of the optical surface is $F(x, y, z) = 0$. $F$ defines a mapping: $F : \gamma \to R$. $\gamma$ is a vector space, $R$ is a set of real numbers, and $F$ is a mapping operator. For the position vector of any point on the optical surface $\mathbf{r}' \in \gamma$, $F(\mathbf{r}') = 0$. Therefore, the intersection of light and the optical surface can be obtained by the equation $F(t\mathbf{a} + \mathbf{b}) = 0$. The equations of the intersections of multiple rays and the same optical surface are independent of each other. The intersections of the same light and different optical surfaces are calculated sequentially, that is, unidirectional coupling. This method does not depend on the number, position and shape of the optical surface in the system and has a wide application range.

As shown in Figure 2, the coordinate origin coincides with the vertex of the mirror. The Z axis is the optical axis, A is the ideal surface of the mirror, B is the deformed mirror

surface, C is the actual wavefront passing through $p_5$, D is the Gaussian sphere and E is the ideal image surface. D takes the ideal image point $p^*$ as the center of the sphere and $p^*p_5$ as the radius. The direction of incident light is parallel to the optical axis. $\mathbf{n}_1$ is the normal direction at $p_1$. The light starts from $p_0$ and reaches $p_4$ on the image plane through $p_1p_2p_3$. The optical path of $p_2p_3$ is the optical path difference between the actual and the ideal wavefront. To improve the imaging quality of the system, the difference between the actual and the ideal wavefront should be reduced.

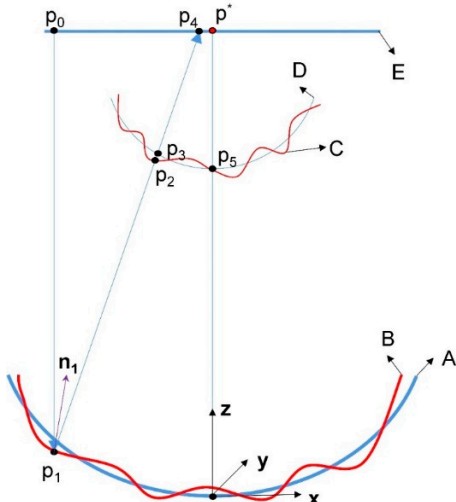

**Figure 2.** Schematic representation of light propagation in a mirror.

Wavefront aberration can comprehensively reflect the imaging performance of the system. The ideal image point of the main light is taken as the Gaussian point. The gaussian sphere is the sphere centered at the Gaussian point with an appropriate radius. Ray tracing can obtain the optical path of the light reaching the Gaussian sphere. The difference between this optical path and the actual optical path to $p_5$ is wavefront aberration. Similar to Zernike fitting of mirror surface deformation, the wavefront aberration coefficients can be obtained by fitting the optical path difference. Since the optical path of the actual wavefront is not easy to solve, the ideal optical path to the Gaussian sphere is used instead. By optimizing the wavefront aberration, the actual wavefront is close to the ideal one, and the imaging performance is improved.

### 2.5. Topology Optimization Model

According to the SIMP method, the mirror is discretized by elements, and the relative density of each element node is used as the design variable to describe the topology of the structure. The material of the elements is a porous material based on the selected material, and the properties of the material can be expressed as follows:

$$E_i = \left( \rho_{\min} + (1 - \rho_{\min}) \rho_i^P \right) E_0 \quad i = 1, 2, \cdots, \widetilde{n} \tag{9}$$

where $E_0$ is the elastic modulus of the selected material, $P$ is the penalty factor, and $\widetilde{n}$ is the total number of grid nodes, $\rho_i$ and $E_i$ are the relative density and elastic modulus of the material at the node $i$, respectively. $\rho_i$ takes 1 or 0 to indicate that there is the selected material or no material at this point. To avoid the singularity of the stiffness matrix $\rho_{\min}$ usually takes 0.001. The topology of the structure can be described by the relative density of the material. Design variables can be expressed as:

$$\boldsymbol{\rho} = (\rho_1, \rho_2, \cdots, \rho_{\widetilde{n}})^T \tag{10}$$

To establish the mathematical model of opto-mechanical coupling, it is necessary to determine the expression of ray tracing in the model. According to the Galois theory [26],

equations higher than fourth order have no analytical solution. Generally, the intersections of the rays with optical surfaces can only be solved numerically. To establish the relationship between ray tracing and design variables, the intersection equations are directly introduced into the optimization model. The mathematical model of topological optimization for opto-mechanical coupling is as follows:

$$obj : J = g(\mathbf{C}_2) \tag{11}$$

$$st : \mathbf{KU} = \mathbf{F} \tag{12}$$

$$\mathbf{DT} = \mathbf{G} \tag{13}$$

$$h_1(\mathbf{T}) \leq T^* \tag{14}$$

$$\mathbf{H}_1\mathbf{C}_1 = \mathbf{P}_1 \tag{15}$$

$$\begin{cases} S_1(k_{1i}t_{1i} + k_{0i}) - \sum\limits_{j=1}^{\tilde{a}} C_{1j}\varphi_{1j}(k_{1i}t_{1i} + k_{0i}) = 0 \\ S_2(k_{2i}t_{2i} + k_{1i}t_{1i} + k_{0i}) = 0 \\ S_3(k_{3i}t_{3i} + k_{2i}t_{2i} + k_{1i}t_{1i} + k_{0i}) = 0 \end{cases} \quad i = 1, 2, \cdots, L \tag{16}$$

$$\mathbf{H}_2\mathbf{C}_2 = \mathbf{P}_2 \tag{17}$$

$$h_2(\mathbf{C}_1) \leq C_1^* \tag{18}$$

$$h_3(\mathbf{C}_2) \leq C_2^* \tag{19}$$

$$\int \rho d\Omega \leq \alpha V_0 \tag{20}$$

$$0 \leq \rho_i \leq 1 \tag{21}$$

$$R(\rho) = 0 \tag{22}$$

Equation (11) is the objective of optimization, which is an analytic function of the wavefront aberration coefficients $\mathbf{C}_2$. Equation (12) is the finite element solution equation of the mirror deformation. $\mathbf{K}$, $\mathbf{U}$ and $\mathbf{F}$ are the system stiffness matrix, the displacement vector and the force vector, respectively. Equation (13) is the equation of rigid body displacement, which can solve the rigid body displacement of six degrees of freedom of the mirror surface. Formula (14) is the constraint of the rigid body displacement, which is the analytical function of the rigid body displacement, and $T^*$ is the upper bound of the constraint. There can be several different rigid body displacement constraints. Equation (15) is the solution equation of mirror deformation fitting. $\mathbf{C}_1$ is the coefficient vector to be solved and $\tilde{a}$ is the number of bases used for mirror deformation fitting.

Formula (16) represents the equations of intersections of rays and the deformed mirror, the Gauss reference sphere and the Gauss image plane, respectively. $S_1$ denotes the operator corresponding to the ideal mirror surface, and $\varphi_{1j}$ denotes the operator corresponding to the $j$th term in the bases. $C_{1j}$ is the $j$th term of the mirror deformation fitting coefficients. $k_{0i}$ is the position vector of the exit point of the $i$th ray, and $t_{1i}$ represents the optical path of the $i$th ray from the exit point to the mirror surface. $S_2$ denotes the operator corresponding to the Gaussian reference sphere. $t_{2i}$ is the optical path of the $i$th ray from the mirror surface to the Gaussian reference sphere. $S_3$ is the operator corresponding to the Gaussian image. $t_{3i}$ is the optical path of the $i$th ray from the Gaussian reference sphere to the image plane. $k_{1i}$, $k_{2i}$ and $k_{3i}$ are the unit vectors of the light direction corresponding to the optical paths $t_{1i}$, $t_{2i}$ and $t_{3i}$, respectively. $L$ is the number of rays. The intersection calculation between the same light and different optical surfaces is unidirectionally coupled, and the intersection calculation equations of different lights are independent of each other. The relation between the optical path and design variables is established, and the influence of mirror deformation on light propagation is also reflected.

Equation (17) represents the fitting of the optical path difference. $\mathbf{C}_2$ is the coefficient vector of optical path difference. Equations (18) and (19) are the constraints of coefficients

$\mathbf{C}_1$, $\mathbf{C}_2$, respectively. $h_2$ and $h_3$ are the analytic functions of $\mathbf{C}_1$, $\mathbf{C}_2$, respectively. $C_1^*$ and $C_2^*$ are the upper bounds of the constraints. Multiple different constraints can be imposed on the coefficients. Formula (20) is the volume constraint. $\alpha$ is the upper limit of volume fraction. The mass of the structure can be controlled by volume. Equations (21) and (22) represent the range and symmetry constraints of the design variables, respectively.

The mathematical model can be adjusted according to the specific application. When the model has multiple mirrors, the fitting equations of rigid body displacement and deformation, and ray tracing equations are added according to the number of mirrors.

### 2.6. Sensitivity Analysis by Adjoint Method

Topology optimization is characterized by a large number of design variables. If the difference method were used to solve the sensitivity, the amount of calculation would be very large. To analyze the sensitivity efficiently, the adjoint method is used. In the analysis of the forward problem, the assembly of the stiffness matrix takes up a lot of time. The adjoint method can reduce the computational burden and improve efficiency by reusing the stiffness matrix. The difference method needs to solve the forward problem and analyze the sensitivity after each variable is perturbed separately. Because of the large number of design variables, the amount of computation is unacceptable.

For the topology optimization of multi-physical fields involving multiple constraints, the stepwise solution is adopted to make the sensitivity analysis clear. The derivation of the matrix adopts a molecular layout. The objective and Equation (19) are analytic functions of wavefront aberration coefficients. Their sensitivity analysis can be performed in the same way, and the sensitivity has the same form. Here, only the sensitivity of the objective is analyzed and explained. According to the topology optimization model of optomechanical coupling, the Lagrange function can be constructed by introducing Lagrange multipliers $\lambda_1$, $\lambda_2$, and $\lambda_3$. Therefore, the sensitivity and adjoint equation also can be obtained as follows.

$$\frac{dg(\mathbf{C}_2)}{d\boldsymbol{\rho}} = -\lambda_1^{\mathbf{T}}(\mathbf{w}_2 \circ \boldsymbol{\varphi}_2)^T(-\lambda_2 \circ \mathbf{A}_5 + \lambda_3 \circ \mathbf{A}_6 \circ \mathbf{t}_2) \tag{23}$$

$$\mathbf{H}_2\lambda_1 = -\left(\frac{dg(\mathbf{C}_2)}{d\mathbf{C}_2}\right)^T \tag{24}$$

$$1 + \lambda_2 \circ (\mathbf{A}_1 - \mathbf{A}_2) + \lambda_3 \circ \mathbf{A}_3 = 0 \tag{25}$$

$$1 + \lambda_3 \circ \mathbf{A}_4 = 0 \tag{26}$$

$$\mathbf{w}_{2\mathbf{j}} = [w_{2j1}, w_{2j2}, \cdots, w_{2jL}]^T \tag{27}$$

$$\mathbf{w}_2 = [\mathbf{w}_{21}, \mathbf{w}_{22}, \cdots, \mathbf{w}_{2\tilde{b}}] \tag{28}$$

$$\boldsymbol{\varphi}_{2\mathbf{j}} = [\varphi_{2j1}, \varphi_{2j2}, \cdots, \varphi_{2jL}]^T \tag{29}$$

$$\boldsymbol{\varphi}_2 = [\boldsymbol{\varphi}_{21}, \boldsymbol{\varphi}_{22}, \cdots, \boldsymbol{\varphi}_{2\tilde{b}}] \tag{30}$$

$$\lambda_2 = [\lambda_{21}, \lambda_{22}, \cdots, \lambda_{2L}]^T \tag{31}$$

$$\lambda_3 = [\lambda_{31}, \lambda_{32}, \cdots, \lambda_{3L}]^T \tag{32}$$

$$\mathbf{t}_2 = [t_{21}, t_{22}, \cdots, t_{2L}]^T \tag{33}$$

$$Q_{1i} = k_{1i}t_{1i} + k_{0i} \quad i = 1, 2, \cdots, L \tag{34}$$

$$Q_{2i} = k_{2i}t_{2i} + k_{1i}t_{1i} + k_{0i} \quad i = 1, 2, \cdots, L \tag{35}$$

$$\mathbf{A}_1 = [\nabla S_1(Q_{11}) \cdot k_{11}, \nabla S_1(Q_{12}) \cdot k_{12}, \cdots, \nabla S_1(Q_{1L}) \cdot k_{1L}]^T \tag{36}$$

$$\mathbf{A}_2 = [\sum_{j=1}^{\tilde{a}} C_{1j}\nabla\varphi_{1j}(Q_{11}) \cdot k_{11}, \sum_{j=1}^{\tilde{a}} C_{1j}\nabla\varphi_{1j}(Q_{12}) \cdot k_{12}, \cdots, \sum_{j=1}^{\tilde{a}} C_{1j}\nabla\varphi_{1j}(Q_{1L}) \cdot k_{1L}]^T \tag{37}$$

$$\mathbf{A}_3 = [\nabla S_2(Q_{21}) \cdot \boldsymbol{k_{11}}, \nabla S_2(Q_{22}) \cdot \boldsymbol{k_{12}}, \cdots, \nabla S_2(Q_{2L}) \cdot \boldsymbol{k_{1L}}]^T \tag{38}$$

$$\mathbf{A}_4 = [\nabla S_2(Q_{21}) \cdot \boldsymbol{k_{21}}, \nabla S_2(Q_{22}) \cdot \boldsymbol{k_{22}}, \cdots, \nabla S_2(Q_{2L}) \cdot \boldsymbol{k_{2L}}]^T \tag{39}$$

$$\mathbf{A}_5 = [\sum_{j=1}^{\tilde{a}} \frac{dC_{1j}}{d\boldsymbol{\rho}} \varphi_{1j}(Q_{11}); \sum_{j=1}^{\tilde{a}} \frac{dC_{1j}}{d\boldsymbol{\rho}} \varphi_{1j}(Q_{12}); \cdots; \sum_{l=1}^{\tilde{a}} \frac{dC_{1j}}{d\boldsymbol{\rho}} \varphi_{1j}(Q_{1L})] \tag{40}$$

$$\mathbf{A}_6 = [\nabla S_2(Q_{21}) \cdot \frac{\mathbf{d}\boldsymbol{k_{21}}}{\mathbf{d}\boldsymbol{\rho}}; \nabla S_2(Q_{22}) \cdot \frac{\mathbf{d}\boldsymbol{k_{22}}}{\mathbf{d}\boldsymbol{\rho}}; \cdots; \nabla S_2(Q_{2L}) \cdot \frac{\mathbf{d}\boldsymbol{k_{2L}}}{\mathbf{d}\boldsymbol{\rho}}] \tag{41}$$

where $\widetilde{b}$ is the number of terms of the orthogonal bases used to fit wavefront aberration. $L$ is the number of rays. $w_{2ji}$ represents the weight of the node on the fitted surface corresponding to the intersection of the $i$th ray and the Gaussian sphere. $\varphi_{2ji}$ is the value of the $j$th term of the bases used to fit wavefront aberration at the intersection of the $i$th ray and the Gaussian sphere.

According to the optimization model, $h_2(\mathbf{C}_1)$ is the analytical function of $\mathbf{C}_1$. The Lagrange multipliers $\lambda_4$ and $\lambda_5$ are introduced to construct the Lagrange function. The sensitivity of $h_2(\mathbf{C}_1)$ is as follows.

$$\frac{dh_2(\mathbf{C}_1)}{d\boldsymbol{\rho}} = -\lambda_4^T \lambda_5^T \left( \frac{d\mathbf{K}}{d\boldsymbol{\rho}} \mathbf{U} - \frac{d\mathbf{F}}{d\boldsymbol{\rho}} \right) \tag{42}$$

$$\mathbf{H}_1 \lambda_4 = -\left( \frac{dh_2(\mathbf{C}_1)}{d\mathbf{C}_1} \right)^T \tag{43}$$

$$\mathbf{K}\lambda_5 = -\left( \frac{d\mathbf{P}_1}{d\mathbf{U}} \right)^T \tag{44}$$

According to the above, $h_1(\mathbf{T})$ is the analytical function of $\mathbf{T}$. In the same way, the Lagrange function can be constructed by introducing the Lagrange multipliers $\lambda_6$ and $\lambda_7$. The following results can be obtained.

$$\frac{dh_1(\mathbf{T})}{d\boldsymbol{\rho}} = -\lambda_6^{\mathbf{T}} \lambda_7^{\mathbf{T}} \left( \frac{d\mathbf{K}}{d\boldsymbol{\rho}} \mathbf{U} - \frac{d\mathbf{F}}{d\boldsymbol{\rho}} \right) \tag{45}$$

$$\mathbf{D}\lambda_6 = -\left( \frac{dh_1(\mathbf{T})}{d\mathbf{T}} \right)^T \tag{46}$$

$$\mathbf{K}\lambda_7 = -\left( \frac{d\mathbf{G}}{d\mathbf{U}} \right)^T \tag{47}$$

### 2.7. Topology Optimization Process

The mirror is designed according to the actual demand, that is, the parameters such as the aperture, thickness, focal length and support position of the mirror are determined. The solid mirror is used as the initial model of optimization. The finite element model of the mirror is established, including adding boundary conditions and specifying materials. The analysis of the forward problem includes the finite element analysis of mirror deformation, mirror deformation fitting, ray tracing and aberration analysis. The optimization model is established by the SIMP method, and the sensitivity is solved according to the derivation. The part with optical or supporting function in the mirror is the non-design domain, and other areas can be used as the design area.

The problem is a non-convex optimization with multiple constraints, which can be solved by the sequential linear programming method, the method of moving asymptotes (MMA) and other algorithms. In this paper, the MMA algorithm [27] is used to solve the optimization problem. In this algorithm, the optimization problem is convex approximated, and the subproblem is solved in each iteration step. The solution of the subproblem is used to approximate the solution of the original problem. Topology optimization is mainly used

in the conceptual design stage to provide a structure with good performance for subsequent design. The optimized structure is extracted by the density truncation method. The specific optimization process is shown in Figure 3.

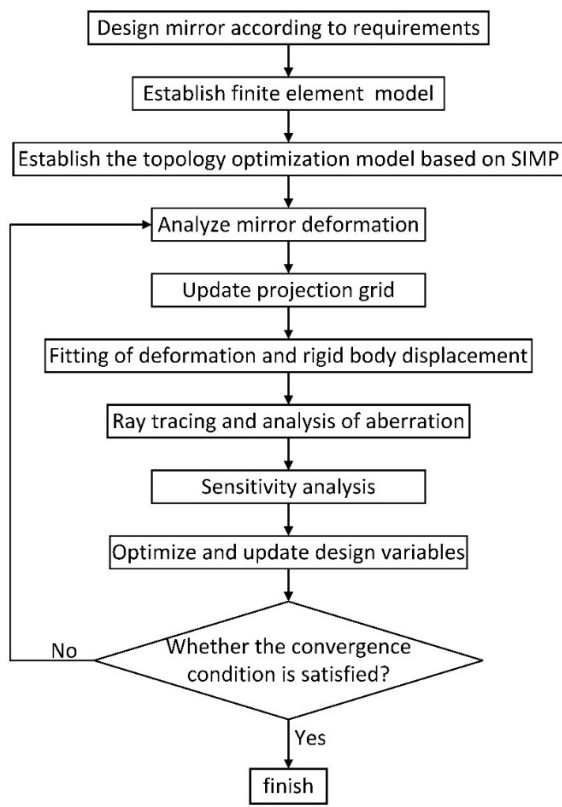

**Figure 3.** Flow chart of optical–mechanical coupling topology optimization.

## 3. Topology Optimization of the Mirrors of the Cassegrain System

Based on the idea of opto-mechanical coupling, the mirror is optimized with wavefront aberration as the objective. The effect of mirror deformation on imaging quality is fully reflected by the optical analysis based on the fitted deformation. The opto-mechanical coupling occurs on the deformed surface. Since the effect of the light energy on the mirror temperature is not considered, the opto-mechanical coupling is unidirectional. The method in this paper is applied to the structural design of the primary and secondary mirrors of the Cassegrain system. The annular primary mirror is optimized in the vertical and horizontal optical axis, and the secondary mirror is optimized in the vertical optical axis. The results show that the method in this paper is effective and applicable to the optimization of different types of mirrors and different working conditions.

### 3.1. Structures and Finite Element Models of the Mirrors

As shown in Figure 4, the Cassegrain system consists of two mirrors. The primary mirror is a parabolic mirror with a central hole, and the secondary mirror is a circular hyperboloid mirror. The primary mirror is supported by a mandrel. The mandrel is a stepped shaft with a center hole, which is connected with the main mirror by glue and the frame by bolts. The secondary mirror is fixed with threaded rings. In the finite element model, the primary mirror and the mandrel are analyzed simultaneously, and the constraints of the degrees of freedom are added to the positions of the bolt holes of the mandrel. The degrees of freedom of the secondary mirror are constrained at three uniform installation positions.

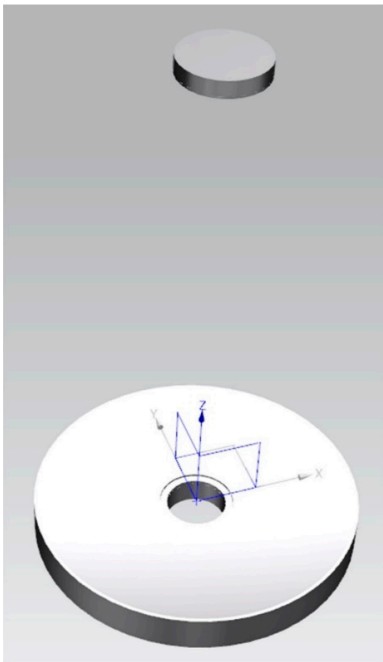

**Figure 4.** The Cassegrain system.

The structure of the annular mirror is shown in Figure 5. The coordinate system is established by taking the mirror surface vertex as the origin. The outer diameter of the mirror is 460 mm. The surface of the mirror is a paraboloid with a focal length of 900 mm, an outer circle diameter of 450 mm, and an inner circle diameter of 104 mm. The height of the inner ring is 2 mm. The center thickness is 40 mm. The thickness of the panel is 8 mm. The diameter of the supporting hole is 84 mm and its wall thickness is 13 mm. The mandrel and the mirror are made of invar steel and SiC material, respectively. SiC material has excellent mechanical properties, thermal properties and good processability. Therefore, it is a good material for mirrors. The panel of the mirror has an optical function, the center ring provides the support position and the mandrel provides the support. These areas are all non-design areas. The remaining part is the design domain, shown in yellow.

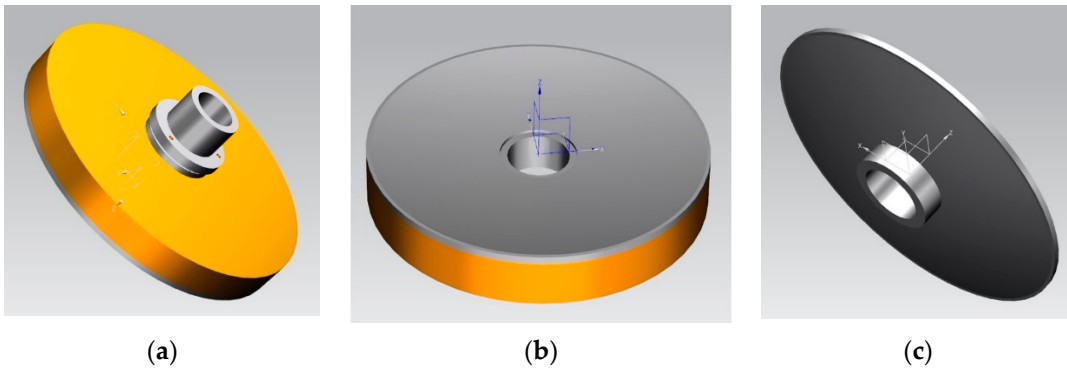

(**a**)    (**b**)    (**c**)

**Figure 5.** The primary mirror of the Cassegrain system. (**a**) The components of the primary mirror; (**b**) The structure of the annular mirror; (**c**) The non-design domain of the annular mirror.

The structure of the secondary mirror is shown in Figure 6. The outer diameter of the mirror is 115 mm. The mirror surface is a hyperboloid with an outer diameter of 105 mm. Panel thickness is 3 mm. The wall thickness of the outer ring and the center thickness of the mirror are 5 mm and 25.573 mm, respectively. The mirror is made of glass–ceramic. The panel and the outer ring are non-design domains, and the remainder is the design domain, shown in yellow.

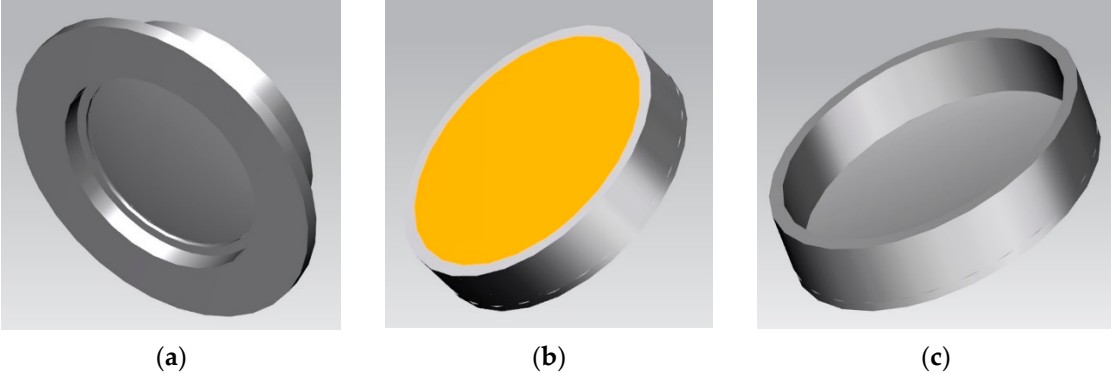

**Figure 6.** The secondary mirror of the Cassegrain system. (**a**) The components of the secondary mirror; (**b**) The structure of the circle mirror; (**c**) The non-design domain of the circle mirror.

As shown in Figure 7, the mirrors are discretized by linear hexahedral elements, and the grids are divided by sweeping in the optical axis direction. The finite element model of the primary mirror has 599,514 elements and 657,450 nodes. The finite element model of the secondary mirror has 95,940 elements and 111,525 nodes. The optimization model uses nodal design variables. To reduce the complexity of the optimization problem, only the gravity of the non-design domain is considered. According to the symmetry of the mirror, the corresponding symmetry constraints are considered in the optimization.

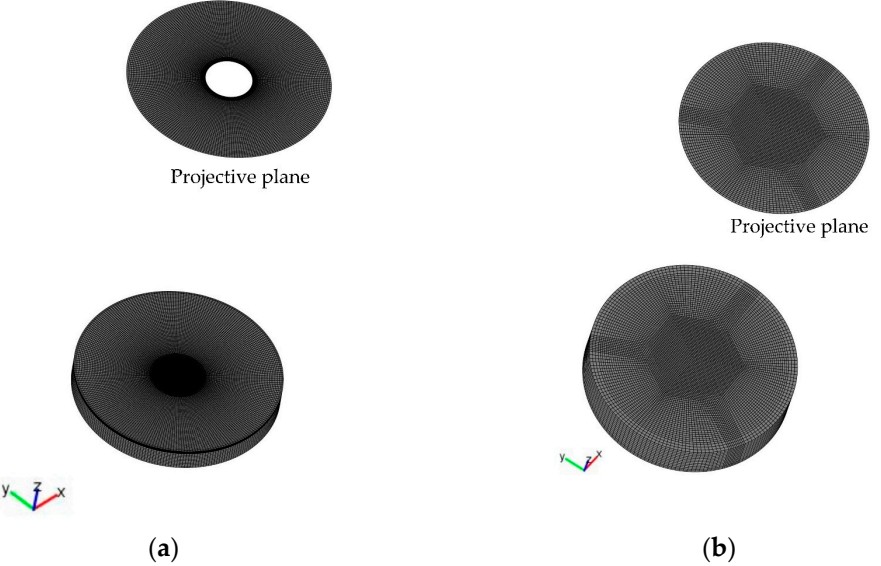

**Figure 7.** Finite element models of the mirrors. (**a**) The primary mirror; (**b**) The secondary mirror.

### 3.2. Optimization Model and Results of the Primary Mirror

The topology optimization model has been explained above, and the model is supplemented here according to the specific application. The aberration optimization of the mirror is a multi-objective optimization problem, which can be transformed into a single objective optimization by a weighted sum. As shown in Equation (48), the weighted sum of the squares of the fitting coefficients of the optical path difference is taken as the objective. The weights are the integral of the square of the bases on the projection plane used for aberration fitting. The objective contains all the aberration coefficients, and the constraints expressed in Equation (19) are not considered here. Different optimization results can be obtained by changing the weight or using different aberration coefficients as objective or

constraints. Therefore, the aberration can be optimized directly and effectively. The sixteen annular Zernike bases are used to fit the mirror deformation and optical path difference.

$$\text{obj} : J = \sum_{j=1}^{\tilde{b}} C_{2j}^2 \int \varphi_{2j}^2 d\Omega \tag{48}$$

$$st : T_x^2 + T_y^2 + T_z^2 + R_x^2 + R_y^2 + R_z^2 \leq T^* \tag{49}$$

$$st : \int \left( \delta - \sum_{j=1}^{\tilde{a}} C_{1j} \varphi_{1j} \right)^2 ds \leq C_1^* \tag{50}$$

Equation (49) is the rigid body displacement constraint of the mirror surface, which is the weighted sum of squares of the rigid body displacement. The constraint of the rigid body displacement can have different weights or schemes. Equation (50) is the constraint of the residual of the fitted mirror deformation. The effect of residual on imaging performance can be reduced and controlled. The optical path diagram of the primary mirror is shown in Figure 8. The Gaussian sphere radius is 0.4 m. The outgoing positions of the rays are uniformly distributed on the ideal image plane, and the incident direction is parallel to the optical axis.

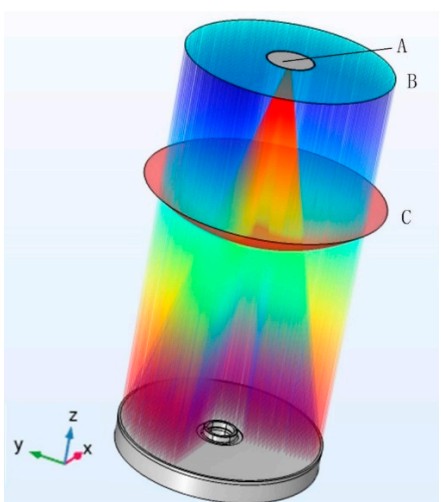

**Figure 8.** Optical path diagram of the primary mirror. A: The ideal image plane; B: The outgoing surface of light; C: The Gaussian reference sphere.

In the case of the vertical optical axis, the upper limit of the normalized rigid body displacement constraint is 0.1, the upper limit of the normalized residual constraint is 0.05, and the upper limit of the volume constraint is 0.2. The initial value of the design variables is 0.2. The mirror assembly has a circumferential one-third cyclic symmetry and is symmetrical about the xz plane. The corresponding symmetry constraints are considered in the optimization. The optimization result of 200 iterations is shown in Figure 9. The structure of the mirror with a density threshold of 0.6 is shown in Figure 10.

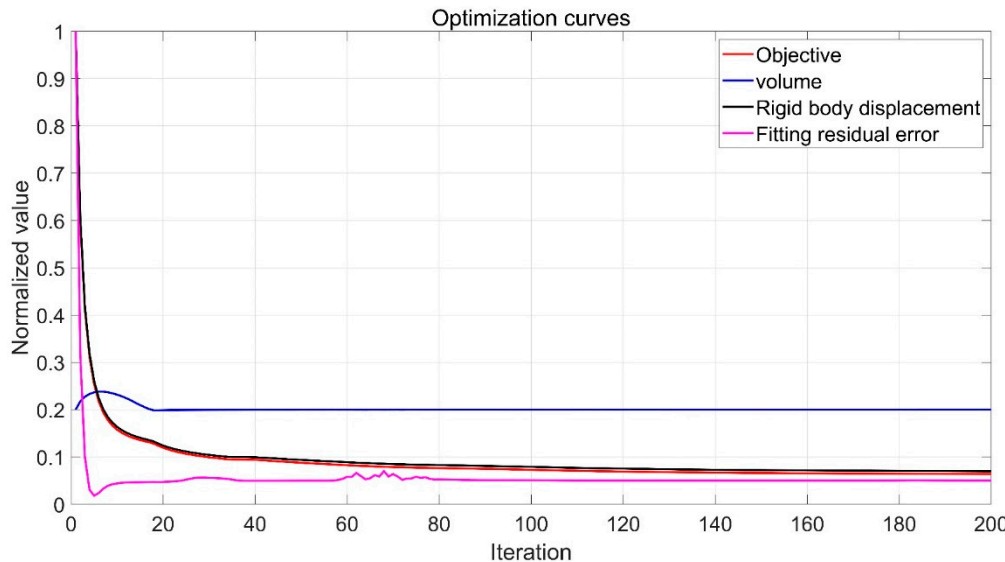

**Figure 9.** Iterative curves of the normalized objective and constraints in the vertical condition of the optical axis.

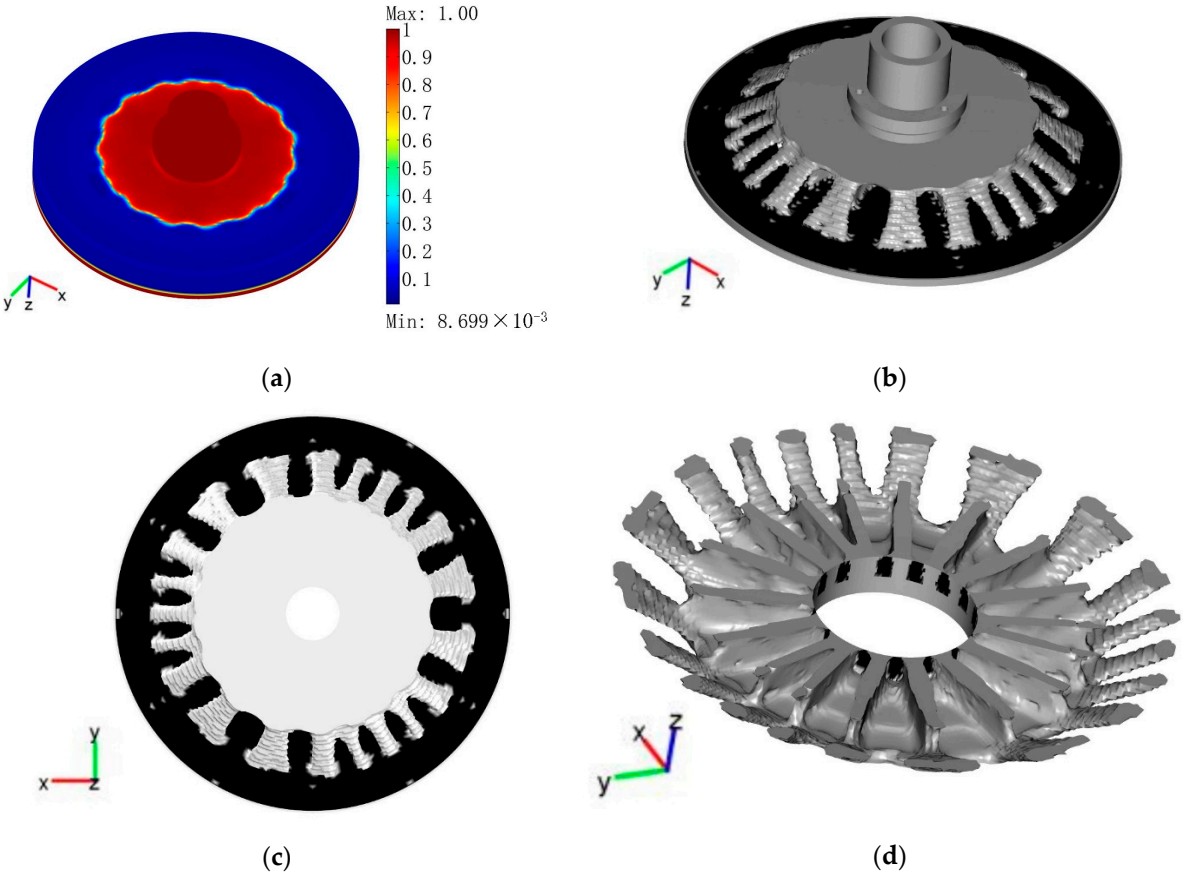

(**a**)

(**b**)

(**c**)

(**d**)

**Figure 10.** Optimization results in the vertical condition of the optical axis. (**a**) The density of the mirror; (**b**) Mirror structure with a density threshold of 0.6; (**c**) Bottom view of the optimized mirror; (**d**) Optimal structure of the design domain.

In the case of the horizontal optical axis, the upper limit of the normalized rigid body displacement constraint is 0.65, the upper limit of the normalized residual constraint is 0.05, and the upper limit of the volume constraint is 0.2. The initial value of the design variables is 0.2. Under this condition, the mirror is symmetrical about the xz plane, and the

corresponding constraints are added to the optimization model. The optimization result of 245 iterations is shown in Figure 11. The structure of the mirror with a density threshold of 0.6 is shown in Figure 12.

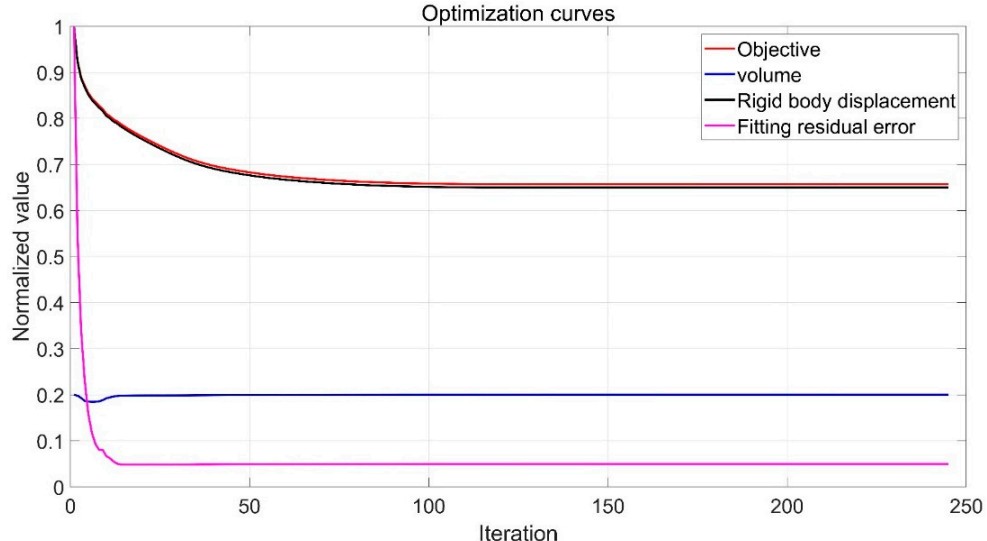

**Figure 11.** Iterative curves of the normalized objective and constraints in the horizontal condition of the optical axis.

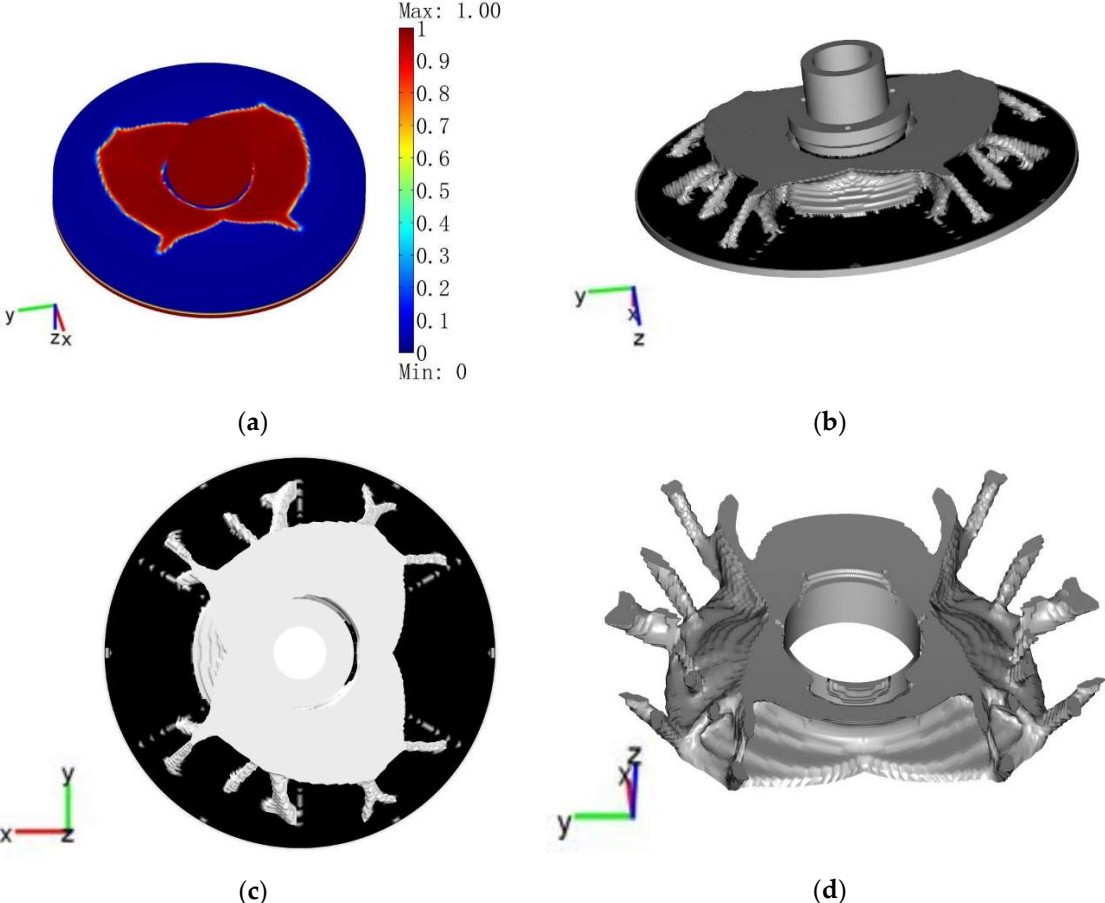

**Figure 12.** Optimization results in the horizontal condition of the optical axis. (**a**) The density of the mirror; (**b**) Mirror structure with a density threshold of 0.6; (**c**) Bottom view of the optimized mirror; (**d**) Optimal structure of the design domain.

It can be seen from the iteration curves of the normalized objective and constraints that all converge, and the constraints are satisfied. To reduce the gray elements in the optimization structure, the number of optimization iterations is increased appropriately.

### 3.3. Optimization Model and Results of the Secondary Mirror

The secondary mirror is a circular mirror with a hyperboloid surface, and the surface deformation is fitted by the 37 fringe Zernike bases. The optical path diagram of the secondary mirror is shown in Figure 13. The outgoing positions of the rays are uniformly distributed on the spherical surface with a radius of 0.9 m and the Gaussian point of the primary mirror as the center. On the projection plane in the direction of the optical axis, the radius of the outer ring and inner ring of the incident points are 0.225 m and 0.052 m, respectively. The aberration is fitted by the 16 annular Zernike bases. The Gaussian sphere is centered at the Gaussian point of the secondary mirror and has a radius of 0.4 m. The topology optimization model of the mirror is the same as that of the primary mirror. The mirror is optimized in the case of the vertical optical axis. The upper limit of the normalized rigid body displacement constraint is 0.15, the upper limit of the normalized residual constraint is 0.1, and the upper limit of the volume constraint is 0.2. The initial value of the design variables is 0.2. The mirror has a one-third circumferential symmetry under this condition, and the corresponding symmetry constraints are added to the optimization model. As shown in Figure 14, the objective and the constraints have converged. Similarly, to reduce the gray elements in the optimized structure, the structure with 200 steps is taken as the optimization result. The optimization results of the secondary mirror are shown in Figure 15.

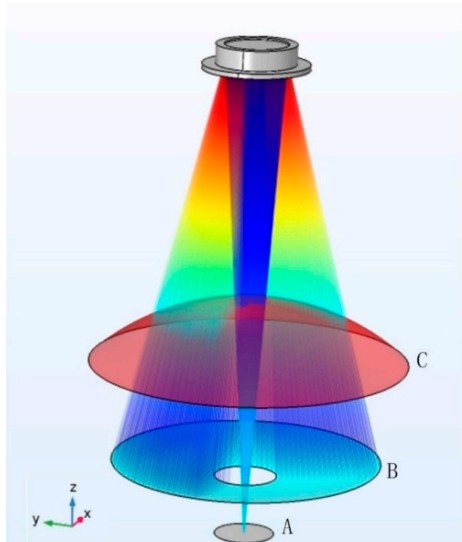

**Figure 13.** Optical path diagram of the secondary mirror. A: The ideal image plane; B: The outgoing surface of light; C: The Gaussian reference sphere.

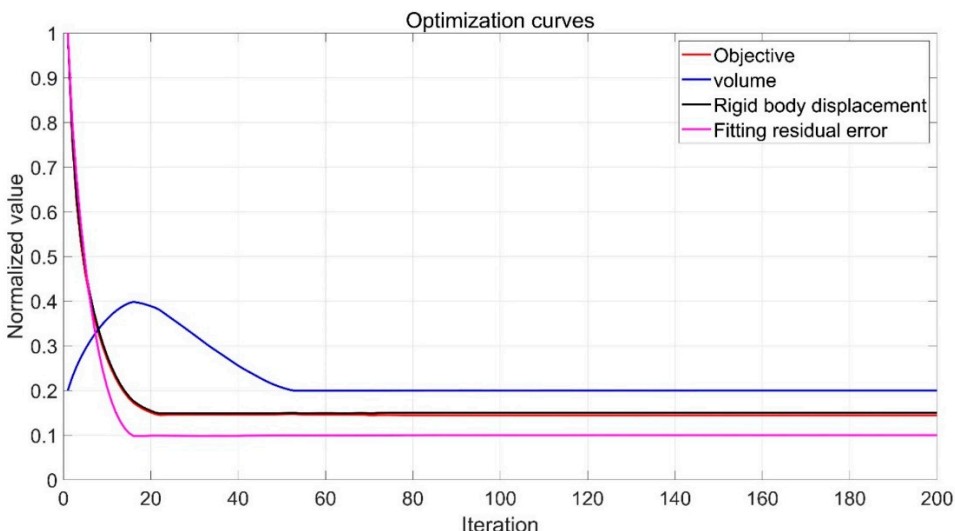

**Figure 14.** Iterative curves of the normalized objective and constraints in the vertical condition of the optical axis.

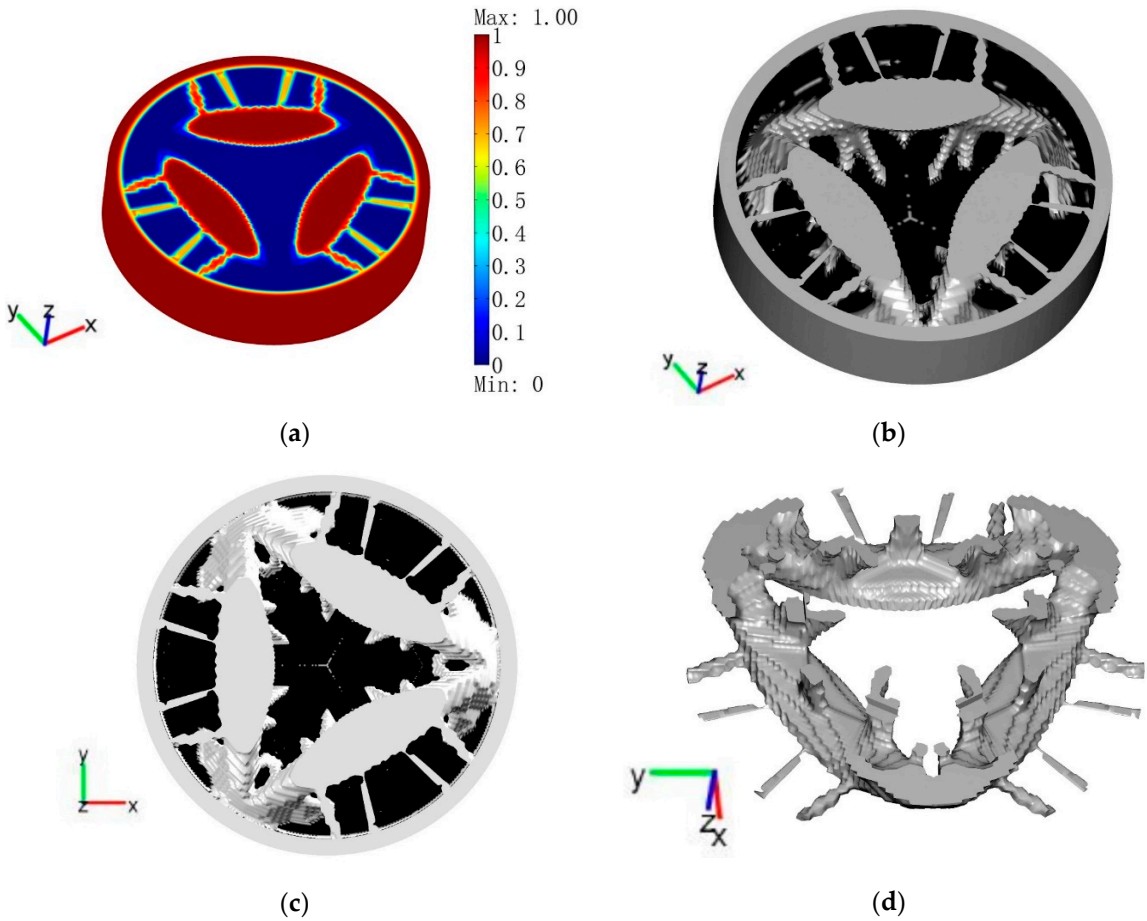

(**a**)

(**b**)

(**c**)

(**d**)

**Figure 15.** Optimization results in the vertical condition of the optical axis. (**a**) The density of the mirror; (**b**) Mirror structure with a density threshold of 0.6; (**c**) Top view of the optimized mirror; (**d**) Optimal structure of the design domain.

## 4. Analysis of the Optimization Results

The optimized structure and the initial structure of the mirror are analyzed by the finite element method under the gravity load. Ray tracing and aberration analysis are performed according to the fitted surface deformation. After removing the deformation

of the first four terms of Zernike fitting, the RMS and the PV of the deformed mirror are calculated. The RMS, PV and aberration objective are shown in Tables 1–3. It can be seen from the tables that the surface accuracy and imaging quality of the optimized mirrors are greatly improved. As shown in Figure 16, the wavefront aberration of the optimized mirrors is obviously reduced.

**Table 1.** Performance comparison between the optimized and initial structure of the primary mirror in the vertical optical axis.

|  | Initial Structure | Optimized Structure | Reduction |
|---|---|---|---|
| RMS (nm) | 14.8032 | 0.8035 | 94.6% |
| PV (nm) | 49.759 | 4.256 | 91.4% |
| Aberration Objective | $5.96 \times 10^{-15}$ | $3.80 \times 10^{-16}$ | 93.6% |

**Table 2.** Performance comparison between the optimized and initial structure of the primary mirror in the horizontal optical axis.

|  | Initial Structure | Optimized Structure | Reduction |
|---|---|---|---|
| RMS (nm) | 6.8972 | 1.9644 | 71.5% |
| PV (nm) | 36.7606 | 10.5474 | 71.3% |
| Aberration Objective | $2.70 \times 10^{-15}$ | $1.77 \times 10^{-15}$ | 34.3% |

**Table 3.** Performance comparison between the optimized and initial structure of the secondary mirror in the vertical optical axis.

|  | Initial Structure | Optimized Structure | Reduction |
|---|---|---|---|
| RMS (nm) | 1.2081 | 0.8378 | 30.7% |
| PV (nm) | 7.121 | 4.989 | 29.9% |
| Aberration Objective | $4.15 \times 10^{-19}$ | $5.98 \times 10^{-20}$ | 85.6% |

In the vertical condition of the optical axis, the Cassegrain system composed of the primary and secondary mirrors optimized in the vertical optical axis is analyzed and compared with the one composed of the initial mirrors. The optical path diagram of the Cassegrain system is shown in Figure 17. The Gaussian sphere is centered at the Gaussian point of the optical system and has a radius of 0.4 m. The weighted sum of squares of the aberration coefficients of the initial structure is $3.691 \times 10^{-16}$, and that of the system composed of the optimized mirrors is $2.256 \times 10^{-17}$. The value of the system composed of the optimized mirrors is reduced by 93.9%. The specific aberration coefficients are shown in Table 4. Except for the 10th and 12th terms of the aberration, the other aberration coefficients of the system are all reduced. As shown in Figure 18, the wavefront aberration of the optimized system is obviously reduced. It can be seen that the method in this paper can effectively improve the imaging quality of the mirror and optical system.

**Table 4.** Wavefront aberration coefficients of the initial and optimized systems in the vertical optical axis.

| Item of Aberration | Initial System (m) | Optimized System (m) | Reduction |
|---|---|---|---|
| Piston | $4.074929 \times 10^{-7}$ | $1.064630 \times 10^{-7}$ | 73.87% |
| Tilt–A | $-4.398294 \times 10^{-12}$ | $-7.236763 \times 10^{-13}$ | 83.55% |
| Tilt–B | $-1.065788 \times 10^{-11}$ | $-5.842915 \times 10^{-12}$ | 45.18% |
| Focus | $1.787158 \times 10^{-7}$ | $2.858994 \times 10^{-8}$ | 84.00% |
| Pri Astigmatism–B | $2.875231 \times 10^{-13}$ | $2.721912 \times 10^{-13}$ | 5.33% |
| Pri Astigmatism–A | $-5.666126 \times 10^{-13}$ | $-1.805944 \times 10^{-13}$ | 68.13% |
| Pri Coma–B | $1.792338 \times 10^{-12}$ | $-6.939779 \times 10^{-14}$ | 96.13% |
| Pri Coma–A | $1.883938 \times 10^{-12}$ | $7.401119 \times 10^{-14}$ | 96.07% |
| Pri Trefoil–A | $2.211011 \times 10^{-9}$ | $1.964020 \times 10^{-9}$ | 11.17% |
| Pri Trefoil–B | $-1.104784 \times 10^{-13}$ | $-1.636671 \times 10^{-13}$ | −48.14% |

**Table 4.** *Cont.*

| Item of Aberration | Initial System (m) | Optimized System (m) | Reduction |
|---|---|---|---|
| Pri Spherical | $-2.977034 \times 10^{-8}$ | $-1.143237 \times 10^{-9}$ | 96.16% |
| Sec Astigmatism–A | $-1.368859 \times 10^{-14}$ | $-4.085603 \times 10^{-14}$ | −198.47% |
| Sec Astigmatism–B | $-5.669197 \times 10^{-14}$ | $-4.603677 \times 10^{-14}$ | 18.79% |
| Sec Coma–A | $-5.161221 \times 10^{-13}$ | $-1.590369 \times 10^{-13}$ | 69.19% |
| Sec Coma–B | $-4.862335 \times 10^{-13}$ | $-1.458645 \times 10^{-13}$ | 70.00% |
| Sec Spherical | $4.971088 \times 10^{-9}$ | $1.226224 \times 10^{-9}$ | 75.33% |

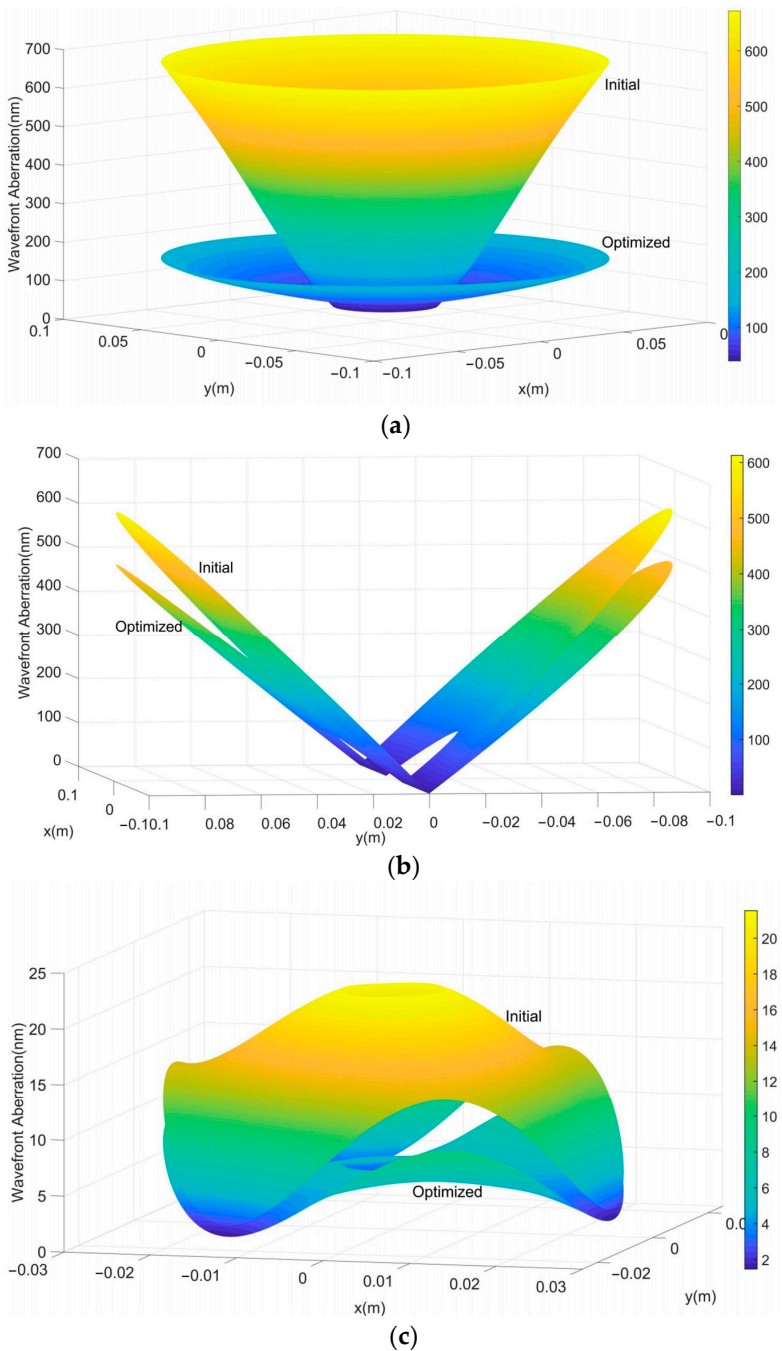

**Figure 16.** Wavefront aberration diagrams of the initial and optimized mirrors. (**a**) The primary mirror in the vertical optical axis; (**b**) The primary mirror in the horizontal optical axis. For clarity, the absolute value of the aberration is taken; (**c**) The secondary mirror in the vertical optical axis.

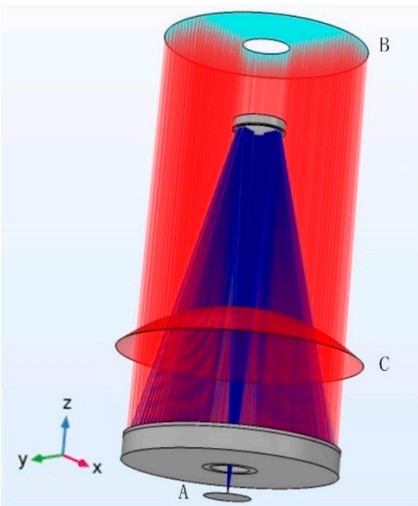

**Figure 17.** Optical path diagram of the Cassegrain system. A: The ideal image plane; B: The outgoing surface of light; C: The Gaussian reference sphere.

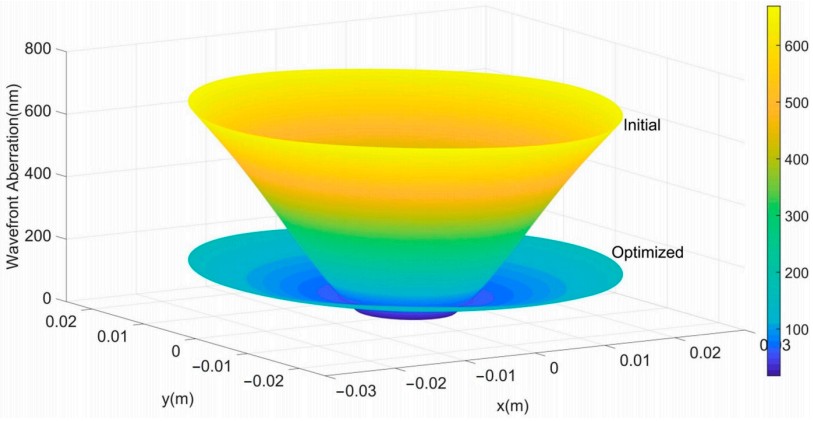

**Figure 18.** Wavefront aberration diagrams of the initial and optimized systems in the vertical optical axis.

## 5. Conclusions

Based on the concept of optical–mechanical coupling, the topology optimization method aiming at the minimization of wavefront aberration is proposed. In the framework of the finite element, the mirror deformation is fitted by orthogonal bases on the projection plane of the mirror surface. Therefore, the deformation data are simplified. The orthogonal bases need to be determined according to the shape of the mirror. The rigid body displacement of the surface deformation is fitted and used as a constraint to reduce its influence on imaging quality during the optimization procedure. The unidirectional opto-mechanical coupling analysis is realized by ray tracing based on the fitted deformation. The functional relationship between optical path difference and design variables is established, which paves the way for the establishment of the mathematical model of topology optimization. The adjoint method is used to analyze the sensitivity efficiently, which reduces the amount of calculation. The effect of the proposed optimization method in this paper is illustrated via the design of mirrors of the Cassegrain system. The optimized structures are extracted and analyzed. Compared with the initial structure, the aberration objective of the primary mirror is reduced by 93.6% in the vertical optical axis and 34.3% in the horizontal optical axis, and the aberration objective is reduced by 85.6% for the secondary mirror in the vertical optical axis. At the same time, the weighted sum of squares of the wavefront aberration coefficients of the Cassegrain system composed of the optimized structures is reduced by 93.9%. This shows that the proposed method is effective and applicable to different working conditions and different types of mirrors, and can improve the imaging

performance of mirrors directly and effectively. In addition, specific aberration coefficients can be optimized by changing their weights in the objective or setting them as constraints.

The research in this paper does not consider the influence of the gravity of the design domain. To further improve the imaging performance of the mirror, the influence of self-weight needs to be further studied. Furthermore, the influence of temperature load can be considered to study the optimization method of reflective mirror from the perspective of optical–mechanical–thermal integration.

**Author Contributions:** Conceptualization, Y.S., W.L. and Z.L.; methodology, Y.S., W.L. and Z.L; software, Z.L.; validation, W.L. and C.W.; investigation, W.L. and Y.T.; resources, Y.Y. and Z.L.; writing—original draft preparation, W.L.; writing—review and editing, W.L. and Z.L.; formal analysis, W.L. and C.W.; visualization, W.L. and Y.T.; supervision, Z.L. All authors have read and agreed to the published version of the manuscript.

**Funding:** This research was funded by the National Science Foundation of China (grant No. 51675506); The Foundation for Excellent Young Scholars of Jilin Province, China (grant number 20190103015JH).

**Institutional Review Board Statement:** Not applicable.

**Informed Consent Statement:** Not applicable.

**Data Availability Statement:** Not applicable.

**Acknowledgments:** We sincerely acknowledge the project's funding support.

**Conflicts of Interest:** The authors declare no conflict of interest.

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
