# Peer review of "Structural Topology Optimization of Reflective Mirror Based on Objective of Wavefront Aberration"

_machines, doi:10.3390/machines10111043_

Round 1
Reviewer 1 Report
The authors present a method to optimize the mirror design for telescopic applications. I find the manuscript interesting. There are a few comments which might help to improve the manuscript.
1. Authors have used the term 'objective' which is confusing to me since the objective is also used as an element in imaging systems and it could also mean the targeted value. Can the authors please specify it clearly?
2. The term 'scheme comparison' is not clear to me. Can authors explain this or provide references?
3. There is no explanation of what CATIA or ANSYS is.
4. Other terms such as 'lightweight rate' are not defined.
Reviewer 2 Report
Wavefront Aberration Objective? -> Please edit the title, there is nothing called "Wavefront Aberration Objective". Maybe ...Wavefront Aberrations of an Objective Lens?
design method of optical-machine structure has always been a hot topic in research. -> always is very long. Please make statement more accurate.
"...carried out with the objective of wavefront aberration" -> please clarify, same problem with wording as in the title.
By analyzing the optimized structure, the aberration objective (remove the word objective) of the primary mirror is reduced by 93.6% in the vertical optical (remove optical) axis and 34.3% in the horizontal optical (remove optical) axis. And the aberration objective (remove the word objective) of the secondary mirror in the vertical optical axis is reduced by 85.6%. At the same time, the aberration objective of the Cassegrain system with optimized structures is reduced by 93.89 %. -> please also use the same number of digits on all numbers, so 93.9%.
limited by the diffraction limit -> limited by diffraction (remove "limit")
The increase of the aperture will cause the increase of the mirror deformation caused by gravity and other factors -> what other factors? Please clarify.
difficulties of directly applying these objectives (remove the word "objectives") to topology
because the mirror surface is usually spatial ? -> it is always spatial, but please edit for clarity.
displacement of the z-direction -> displacement in the z-direction
What are the units of the coefficients shown in Table 4?
4, 11, 13, 22, 26: please edit the text in the references to avoid capital letters only.
In the practical case, the mirror will be affected by temperature changes, how much could that alter the results? Can you model this with Comsol too?
Reviewer 3 Report
Suggestesd new title: Topology Optimization for Reflective Mirrors Based on Solid Isotropic Material with Penalization Model
please align to center figures
when you cite CATIA, ANSYS MATLAB, ISIGHT, please add a note which refer to the official website of the software, a minimun description of the features of the software is reccomended, not mandatory.
